# Viscoelasticity, Mechanical Properties, and In Vitro Bioactivity of Gelatin/Borosilicate Bioactive Glass Nanocomposite Hydrogels as Potential Scaffolds for Bone Regeneration

**DOI:** 10.3390/polym13122014

**Published:** 2021-06-20

**Authors:** Asmaa M. Abd El-Aziz, Ahmed Abd El-Fattah, Azza El-Maghraby, Doaa A. Ghareeb, Sherif Kandil

**Affiliations:** 1Fabrication Technology Research Department, Advanced Technology and New Materials Research Institute, City of Scientific Research and Technological Applications (SRTA-City), Borg Al-Arab, Alexandria 21934, Egypt; maghrabyazza@yahoo.com; 2Department of Materials Science, Institute of Graduate Studies and Research, Alexandria University, El-Shatby, Alexandria 21526, Egypt; a_abdelfattah@alexu.edu.eg (A.A.E.-F.); s.kandil@usa.net (S.K.); 3Department of Chemistry, College of Science, University of Bahrain, Sakhir P.O. Box 32038, Bahrain; 4Biological Screening and Preclinical Trial Laboratory, Department of Biochemistry, Faculty of Science, Alexandria University, Alexandria 21934, Egypt; d.ghareeb@yahoo.com; 5Pharmaceutical and Fermentation Industries Development Centre (PFIDC), City of Scientific Research and Technological Applications (SRTA-City), Borg Al-Arab, Alexandria 21934, Egypt

**Keywords:** hydrogel, gelatin, borosilicate bioactive glass, composite scaffold, viscoelasticity, bone regeneration

## Abstract

Chemical cross-linking was used to create nanocomposite hydrogels made up of gelatin (G) and borosilicate bioactive glass (BBG) with different content (0, 3, and 5 wt.%). The G/BBG nanocomposite hydrogels were studied for their morphology, mechanical properties, and viscoelasticity. SEM images revealed a macroporous interconnected structure with particles scattered across the pore walls. Studies of water absorption and degradation confirmed that the nanocomposite scaffolds were hydrophilic and biodegradable. The addition of 5% BBG to the scaffold formulations increased the compressive modulus by 413% and the compressive intensity by 20%, respectively. At all frequency ranges tested, the storage modulus (G′) was greater than the loss modulus (G″), revealing a self-standing elastic nanocomposite hydrogel. The nanocomposite scaffolds facilitated apatite formation while immersed in simulated body fluid (SBF). According to the findings, G/BBG nanocomposite scaffolds could be a promising biomaterial for bone regeneration.

## 1. Introduction

Bone lesions (such as fractures and tumors) are among the most difficult to treat and manage in medicine. The combination of the high incidence of chronic illnesses, injuries, obesity, and the general rise in the elderly population, necessitates the development and use of more effective treatments for the replacement of injured and impaired tissues. In the operating rooms, auto-, allo-, and xenografts are still the first choice [1,2]. This year (2021) the Transplantation Division (DoT) of the Health Resources and Services Administration (HRSA) is monitoring the effect of the coronavirus (COVID-19) public health emergency on organ donations and transplantation. On behalf of the Board of Directors, the USA Executive Committee has taken a range of steps to help members document COVID-19 problems surrounding organ donation and transplantation and to help members concentrate the necessary resources on critical clinical services (https://optn.transplant.hrsa.gov, accessed on 29 April 2021) in the short-term. Under these conditions, researchers are keen to find new ways to compensate for this organ shortage [3].

The disadvantages of transplant materials such as insufficient harvesting tissue, donor scarcity, and the possibility of disease transmission in modern medicine, hinder their use [3].

Tissue regeneration is currently under intensive review as the principal mechanism involved in cell growth and organ reconstruction. The theory of tissue engineering focuses on the reconstruction of biological structures including neo-functional tissue and/or whole organs. The underlying mechanism that often incorporates both molecular and mechanical signals typically begins with the stimulation of cells through new tissue synthesis [4]. A perfect 3D scaffold not only efficiently stimulates bone regeneration into ideal shapes but also aids in the healing of bone defects [5]. Biomaterials are simply defined as the medium by which these signals are delivered. They are essentially 3D structures with variable concepts and designs that can be modified to provide the therapeutic components necessary for the regulation of living tissue functions or the regeneration of damaged tissue. A scaffold is a natural or synthetic support used to design a biological substitute that, due to superior mechanical and structural properties, aims to provide a higher degree of satisfactory performance than the damaged tissue [6].

Hydrogels, like the extracellular matrix in nature, are porous, soft materials made up of natural and/or synthetic polymer networks with high water content. Furthermore, the flexibility, low toxicity, biodegradability, and easily modifiable networks of the bioactive glass, particularly when combined with biomolecules with regenerative property connections, make them very appealing for bone tissue engineering [7]. Synthetic polymers allow for the production of customised scaffolds, but they have disadvantages, such as the possibility of rejection due to low bioactivity. Alternatively, naturally occurring polymers are biologically active, biocompatible, improve cell adhesion and growth, and have superior biodegradability.

As a biomimetic strategy to mineralize biological tissues and tailor mechanical properties such as hardness and resilience, naturally occurring polymers have been mixed with bioactive glasses and glass-ceramics [8]. Furthermore, borosilicate glass (BBG) has low chemical durability and quickly converts to hydroxyapatite (HA) in physiological media, and bonds directly to bone in a way comparable to silicate-based bioglass. Because of the observed beneficial action of the B ion, and because these glasses can be heat-treated (sintered) at lower temperatures than silicate bioglass and do not crystallise, boron-containing silicate glasses are currently being researched for their possible applications.BBG was obtained from silicate by substituting B_2_O_3_ for the molar concentration of SiO_2_ [9,10].

As a result, our current study aims to develop a new simple sol-gel processing method for the synthesis and the characterization of BBG, and then fabricate polymeric hydrogel nanocomposite scaffolds. The lyophilization process was used to create the polymer–glass scaffolds which are gaining popularity due to their appealing properties. Finally, we examine the properties of the scaffolds, including viscoelasticity, mechanical properties, microstructure, biodegradability, and in vitro bioactivity. To our knowledge, we are the first to prepare a gelatin(G)/BBG hydrogel nanocomposite for bone tissue engineering as a potential application.

## 2. Materials and Methods

### 2.1. Materials

Gelatin from bovine skin (type B) and bovine serum albumin was purchased from Sigma Aldrisch Chemical Co., St. Louis, MO, USA. Tetraethyl orthosilicate (TEOS), calcium nitrate tetrahydrate Ca(NO_3_)_2_∙4H_2_O, triethyl phosphate (C_2_H_5_)_3_PO_4_, citric acid (99%), ammonia solution (25%), and absolute ethanol (98%) were all purchased from Sigma Aldrich Chemical Co., Darmstadt, Germany. Tetraethyl orthosilicate (TEOS) 98% was purchased from Alfa Aesar, Kandel, Germany. Calcium nitrate (Ca (NO_3_)_2_·4H_2_O) was purchased from Belami Fire Chemicals, Mumbai, India. Boric acid (H_3_BO_3_) from Spectrum Chemicals was produced in New Brunswick, Canada. Tri-ethyl phosphate (TEP) and nitric acid (HNO_3_) were produced by Fischer Scientific, Loughborough, UK. Ammonium solution (NH_4_OH) 33% was purchased from El-Nasr Pharmaceutical Chemicals Company, Oubour, Egypt.

### 2.2. Synthesis of BBG-NPs

The BBG-NPs, with a mole composition of 55% SiO_2_, 24% CaO, 6% P_2_O_5_, and 15% B_2_O_3_ were synthesized using the sol-gel process described by Xie et al. [11].

In the synthesis process, TEOS (27.7 mL) and TEP (1.7 mL) were dissolved in a mixture of deionized water (16.88 mL) and absolute ethanol (100 mL), followed by the addition of HNO_3_ solution (2 M, 3.4 mL) to initiate the hydrolysis reaction of TEOS. After stirring for 3 h, H_3_BO_4_ (6.9 g) and Ca(NO_3_)_2_∙4H_2_O (11.214 g) were added sequentially to the above mixture and were then stirred for an additional 2 h. Subsequently, ammonia solution (2 M, 30 mL) was then added dropwise to the obtained sol under continuous stirring for 30 min to form a homogeneous gel. The resulting gel was extracted, washed three times with deionized water, and dried for one day at 80 °C before being calcined at 700 °C for 2 h at a heating rate of 3 °C/min. BBG-NPs were ground without sieving in a mortar and pestle to obtain a fine powder for testing and characterization.

### 2.3. Preparation of G/BBG Nanocomposite Hydrogel

Gelatin solution (5 *w*/*v*%) was made by dissolving gelatin (0.5 g) in deionized water (10 mL) and stirring it for 2 h at 50 °C. After that, ultrasonic vibration was used to suspend 3 wt.% and 5 wt.% of synthesized BBG-NPs in the solution for 30 min. After stirring for 2 h, the bioglasses containing hydrogels were cross-linked with 5 µL·mL^−1^ GA and stirred for 1 h at room temperature, as shown in Figure 1. The cross-linked samples were repeatedly washed with DI water to remove the remnant GA. Prior to characterization, the obtained nanocomposite hydrogels were cooled to the temperature of −25 °C overnight and lyophilized with liquid nitrogen in a freeze drier for 48 h to obtain the interconnected porous hydrogels.

### 2.4. Characterization of BBG-NPs

The morphology and grain size of BBG-NPs were analyzed using a transmission electron microscope (TEM) (model JEOL-JEM-100CX, Tokyo, Japan) operating at an accelerating voltage of 80 kV. Before TEM imaging, a few drops of BBG-NPS (dispersed in ethanol) were placed on carbon-coated copper grids and double-stained for 20 min with freshly prepared uranyl acetate, and then for 2–5 min with lead citrate.

The Fourier transform infrared spectroscopy (FTIR) spectrum for the prepared BBG-NPs was recorded throughout the wavenumber range from 4000 to 400 cm^−1^, at a resolution of 4 cm^−1^ using the Perkin-Elmer FTIR spectrometer. The sample was prepared using the KBr disc technique in which the powder sample (5 mg) was compressed with KBr (200 mg) under hydraulic pressure to form a disk with a diameter of 13 mm.

### 2.5. Characterization of G/BBG Nanocomposite Hydrogels

#### 2.5.1. Scanning Electron Microscopy (SEM)

Scanning electron microscopy (SEM) (JEOL 5300-JSM, Tokyo, Japan) was used to investigate the microstructure and surface morphology of the G/BBG nanocomposite hydrogels. Prior to SEM imaging, the freeze-dried samples were sputter-coated with gold to a thickness of 400 in a sputter-coating unit.

#### 2.5.2. Mechanical Testing

A mechanical measuring system was used to determine the compressive modulus and strength of the G/BBG nanocomposite hydrogels (DMA 7e—Perkin Elmer, Waltham, MA, USA). The cylindrical samples were manufactured with a diameter of 1.5 cm and a thickness of 1 cm. At a strain rate of 0.5 mm/min, a uniaxial compression test was performed. The compressive modulus was determined using the average slope of the strain–stress curve’s initial linear portion (10–20 percent).

#### 2.5.3. Viscoelastic Measurements

A strain-controlled rheometer (Anton Paar Rheoplus V3.62—MCR 302, Graz, Austria) with a temperature sensor was used to test the rheological properties of the control gelatin hydrogel and the G/BBG nanocomposite hydrogel samples. At 37 °C, a dynamic frequency sweep test was performed with a strain amplitude of γ_0_ = (1%), and an angular frequency range of from 1 rad/s to 100 rad/s. As a function of angular frequency (ω), the storage or elastic modulus (G′), the loss or viscous modulus (G′′), the complex shear modulus (G*), as well as the phase shift angle (δ), were measured.

#### 2.5.4. Water Absorption

Samples (*n* = 6) were equilibrated in PBS or deionized water (DI) at room temperature to test the swelling activity of the composite hydrogels. At predetermined times, short intervals (5, 10, 15, 30, and 60 min) and long intervals (6, 12, and 24 h), the samples were removed from the water, wiped down, and weighed to determine their wet weights (Ww). They were then dried in an oven at 40 °C until they reached a consistent weight and then weighed once more to assess their dry weights (Wd). Equation (1) was used to measure the percentage of swelling ratio in each sample [12].
(1)$$\mathrm{Swelling}\;\mathrm{ratio}\;\%=\left[\frac{\mathrm{Ww}-\mathrm{Wd}}{\mathrm{Wd}}\right]\times 100$$

#### 2.5.5. Degradation Studies

Immersion in PBS (pH = 7.4) at 37 °C was used to study the samples’ in vitro degradation. In sterile glass test tubes, pre-weighted (Wi) dry samples were soaked in PBS (5 mL) and incubated for 14 days. At 1, 7, and 14 days, the samples were removed from the solution, washed with deionized water, dried at 40 °C until the mass remained unchanged, and measured. Finally, the samples’ dry weight (Wd) was recalculated, and the percentage of biodegradation was measured using Equation (2) [13].
(2)$$\mathrm{Degradation}\;\%=\left[\frac{\mathrm{Wi}-\mathrm{Wd}}{\mathrm{Wd}}\right]\times 100$$

#### 2.5.6. Bioactivity Evaluation

The preparation of 1 L of SBF, in order, is shown in Table 1. If necessary, the pH was adjusted using hydrochloric acid or sodium hydroxide [14].

The bioactivity of the G/BBG nanocomposite scaffolds was assessed in vitro in simulated body fluid (SBF) at 37 °C as described in previous studies [15,16]. The scaffold sample (0.5 mg) was immersed in SBF (1 mL) and immersion times of up to 14 days were used. At selected times (7 and 14 days) the scaffolds were removed from the SBF, washed twice with deionized water and then twice with ethanol, and dried at 40 °C.

The crystalline phase formed on the surface of the scaffolds by conversion in SBF was analyzed using X-ray diffraction (XRD) measurements (model, XRD-7000, Shimadzu, Japan). Cu-K radiation (λ = 0.1540 nm) was used in the X-ray laser, which was controlled at 40 kV and 38 mA. With a scanning rate of 5° per minute, the XRD pattern was observed in the 2 range from 5° to 80°. The crystalline phase of the nHA particles was determined and compared to reports in the literature as well as the Joint Committee on Powder Diffraction Standards (JCPDS) for HA (PDF number: 74-0566). SEM was used to analyse the surface morphology and microstructure of the scaffolds after immersion in SBF under the previous conditions.

### 2.6. Statistical Analysis

The significant differences between the data among the experimental groups were analysed using a one-way ANOVA test followed by the Turkey’s Post Hoc test to analyze the differences among the experimental groups. All data were presented as standard deviations (mean ± SD) and each experiment was replicated at least five times. A *p* value < 0.05 was considered significant.

## 3. Results and Discussion

### 3.1. Characterization of BBG-NPs

The characterization results of the synthesized BBG-NPs are displayed in Figure 1. As shown in Figure 2a, the FTIR spectrum shows all characteristic bands of BBG. A typical absorption peak at 1150 cm^−1^ corresponds to Si–O–Si asymmetric stretching vibrations [17]. The weak absorption peak at 812 cm^−1^ is assigned to Si–O–Ca or non-bridging oxygen. In addition, the absorption peak at 750 cm^−1^ is attributed to the bending vibration of OPO. The absorption peak at 1636 cm^−1^ is due to the stretching vibration mode of Si-OH with the water molecules. Moreover, the absorption peaks at 703 and 1422 cm^−1^ are related to the bending of BOB linkages in the borate network and B–O symmetric stretching vibrations of the boroxoal ring of BO_3_^3−^ units, respectively [18]. The appearance of peaks from BO_3_^3−^ indicates that BO_3_^3−^ partially substitutes SiO_4_^4−^ sites in the glass network structure. According to the TEM micrograph, as shown in Figure 2b, the synthesized BBG-NPs have nearly sphere-shaped morphology and the particle sizes are less than 50 nm.

### 3.2. Characterization of G/BBG Nanocomposite Hydrogels

Figure 3 shows the cross-linking reaction; the aldehyde (–CHO) of GA can interact chemically with amino groups (–NH_2_) on gelatin to form chemical bonds. Since it is bi-functional, it links two different gelatin molecules, hence its mode of cross-linking [14]. Physical or chemical interactions by non-bonding oxygen (NBO) atoms may be responsible for the integration of the particles of borate bioactive glass (BBG) in the cross-linked hydrogel in both cross-linking methods.

The cross-linking density of the hydrogels was related to the solubility ratio of the hydrogels, determined by immersion in distilled water at room temperature. After different time intervals, the soluble or unreacted part of the samples was removed from water. The samples were weighed after removing the excess water from the surfaces by using blotting paper. The solubility ratio was calculated by using the weight of the hydrogel after extraction (G) and the initial weight of the hydrogel (Gi) and expressed as a percentage [5].
(3)$$\mathrm{Crosslinking}\;\%=\left(\left(\frac{G}{\mathrm{Gi}}\right)*100\right)$$

As shown in Figure 4, the solubility ratio equilibrated at 7.5 percent after 48 h, indicating that the cross-linking proportion was high, at around 93 percent. In comparison to the unmodified samples, the modified samples with borosilicate nanoparticles were cross-linked, affecting both swelling and solubility ratios.

#### 3.2.1. Micrographic Morphology

The SEM micrographs of the control gelatin hydrogel and the G/BBG nanocomposite hydrogels with different BBG contents (3 and 5 wt.%) are shown in Figure 5. All samples revealed macroporous interconnected structures. However, as the BBG content was increased, the G/BBG nanocomposites exhibited rough and porous network structures with particles scattered in the pore walls, which increased in number and occupied a larger region [19]. The hydrogel with BBG nanoparticles may have better mechanical properties than pure gelatin hydrogel, implying a better microenvironment for cell attachment and development [19].

#### 3.2.2. Mechanical Properties

The compressive modulus and compressive strength of the control gelatin hydrogel and G/BBG nanocomposite hydrogels are summarized in Table 2. In general, both the compressive modulus and compressive strength of the pure gelatin hydrogel increased with the increase in the BBG content. For example, the addition of 5 wt.% BBG noticeably increased the compressive modulus of the gelatin hydrogel from 33.84 to 173.63 KPa (413%) and compressive strength from 12.20 to 14.75 Kpa (20%) in the nanocomposite hydrogel. Furthermore, by the end of the compression test, all of the hydrogel samples had maintained a flat sheet shape and did not spatter randomly.

These findings can be explained by the effect of uniformly distributed BBG nanoparticles that can pass loads from the polymer chain to the particles and prevent microcrack formation [12]. Furthermore, the heavy interfacial bonding between the gelatin chains and the BBG nanoparticles contributes to the additional strength of the nanocomposites [19]. Compared to other reported nanocomposite hydrogels [20,21,22], the G/BBG nanocomposite hydrogels have markedly superior mechanical properties as bone scaffolds for load-bearing applications.

#### 3.2.3. Viscoelastic Properties

A nanocomposite hydrogel scaffold can act as a low-viscosity liquid during application and then as an elastic solid after filling the defect, with a fast transition between the two states [23].

The variation in viscoelastic properties of the nanocomposite hydrogels was characterized by rheological experiments. Rheological tests were used to classify the differences in viscoelastic properties of the nanocomposite hydrogels. Figure 6 and Figure 7 show the relationship between the nanocomposite hydrogels’ storage (G′), loss (G′′), and complex shear (G*) moduli, as well as the phase shift angle (δ).

First, as shown in Figure 6, the G″ values were less than the G′ values over the frequency range studied for all of the hydrogel samples, which confirms that the hydrogels are elastic rather than fluid-like in nature [24]. There was no meaningful change in G′ or G″ values with an increasing angular frequency (1–200 rad/s). In addition, both G′ and G″ values increased with increasing BBG content, which signifies good interfacial compatibility between the gelatin chains and the BBG nanoparticles [20]. Secondly, as shown in Figure 7, the G* values did not improve with increasing angular frequency for the nanocomposite hydrogels with low BBG content (3 wt.%), but there was a substantial increase in this property with high BBG content (5 wt.%). The G* values of the nanocomposite hydrogel reinforced with 5 wt.% BBG were 65 ± 1.06 KPa at 25 rad/s, 75 ± 1.71 KPa at 100 rad/s, and 85 ± 1.42 KPa at 200 rad/s, respectively, which were consistent with previously published G* values for cancellous bone scaffolds [24]. Finally, it is accepted that for simply elastic materials δ = 0°, and for pure viscous materials δ = 90°, values between 0° and 45° may be appropriate for viscoelastic materials [21]. The values for the δ were below 45° (Figure 7B), suggesting that the hydrogels revealed a more solid-like nature.

This implies the introduction of BBG-NPs as being both physically entrapped and chemically bound to the hydrogel network via the hydrogen bonds. Consequently, the nanocomposite hydrogel networks exhibited improved structural rigidity and reduced elasticity [25]. Moreover, the nanocomposite hydrogels can still flow if subjected to an external shear, which breaks the polymer networks. However, when the shear is removed, the polymer networks will rearrange again and form the hydrogel. This means that the hydrogel could be introduced into any shape, maintaining its molecular structure and the final mechanical properties.

#### 3.2.4. Swelling Behavior

During in vitro cell culture studies, the swelling behavior of the hydrogel scaffolds is significant. The swelling of the hydrogel scaffolds allows for body fluid to be absorbed and for cell nutrients and metabolites to be transferred within the scaffolds. Furthermore, swelling expands the pore size of the scaffolds, increasing the internal surface area available for cell infusion and attachment. Swelling under physiological conditions should be controlled, otherwise the bone scaffolds can weaken and degrade quickly. Figure 8 depicts the swelling of the control and the nanocomposite scaffolds as a result of water absorption in DI at room temperature. Due to their highly interconnected and porous nature, all three classes of scaffolds were able to swell by absorbing water and showed relatively high water absorption [22]. The swelling ratio of a scaffold containing 0 wt.% BBG, alternatively, was substantially higher than that of the scaffolds containing different quantities of BBG. After 1 h, the swelling caused a mass increase of 130 ± 3.41% in the control scaffold, while the swelling of the scaffold containing 5 wt.% BBG was limited to 80 ± 2.71%, that is a 62.5% reduction in the swelling ratio. The swelling behavior of the hydrogels is affected by the hydrophilic group present in the polymer chains [6]. The swelling ratios were equilibrated after 30 min as shown in the following Figure 8A. This phenomenon may be explained by the physical trapping of the BBG-NPs within the hydrogel, which is caused by hydrogen bonds forming between the BBG surfaces and the polymer matrix. The G/BBG scaffolds were supposed to become stronger as the water absorption was reduced. Furthermore, since the nanocomposite scaffolds are hydrophilic, they can be used for bone regeneration. The hydrophilicity of the scaffolds allows for the absorption of body fluid, which primarily consists of water, and is critical for nutrient and metabolite diffusion.

#### 3.2.5. In Vitro Degradation Behavior

The physiological stability of composite hydrogels is crucial during bone regeneration; the degradation rate should correspond to bone ingrowth and increase functional recovery [19]. The degradation of the control and nanocomposite scaffolds was studied in PBS solution for 14 days. As shown in Figure 9, the incorporation of BBG in the scaffolds revealed a significant decrease in the degradation rate. It can be observed that the degradation rates of the scaffolds were dependent on the BBG contents. For example, the control scaffold showed a degradation rate of 14.5%, 27.8%, and 34.2% on the first, seventh, and fourteenth days, respectively. On the first, seventh, and fourteenth days, the 5 wt.% BBG nanocomposite scaffold showed a lagging degradation rate of 8.3 percent, 14.7 percent, and 17.3 percent, respectively.

This may be due to the interactions between the polymer and the BBG-NPs forming a more stable and compact scaffold structure. The reduced degradation result is consistent with other studies in which BBG-based scaffolds were found to have less degradation [26]. The scaffold’s slower biodegradation rate can be advantageous because it helps the scaffold to retain its mechanical integrity while allowing enough time for bone growth to occur in the implant.

#### 3.2.6. Apatite-Forming Ability

Apatite formation on the surface of the hydrogel has been shown to promote osteoblast proliferation and differentiation in previous studies. The ability to shape apatite can also be used to assess osteogenic bioactivity in vitro [27].

Figure 10a shows SEM images of the control and 5 wt.%BBG nanocomposite scaffolds after 14 days of incubation in SBF. The surface of the 0 percent BG scaffold was smooth and apatite-free after 14 days of incubation. On the surfaces of the scaffold containing 5 wt.% BBG, however, flake-like apatite and flower-like crystals were visible (Figure 10b). XRD analysis revealed the formation of crystalline hydroxyapatite (HA) on the surfaces of the nanocomposite scaffold after 14 days of incubation in SBF. In nanocomposite containing 5 wt.% BBG, XRD patterns (Figure 10c) revealed new peaks at 2Θ = 26° (002), 32° (211), 39° (310), 46° (222), 49° (213), and 55° (004), which were consistent with the crystallinity for HA results (JCPDS-PDF no. 74-0566). The G/BBG nanocomposites hydrogels could induce HA formation in SBF, according to SEM and XRD results.

## 4. Conclusions

Chemical cross-linking was used to successfully synthesize nanocomposite hydrogels from BBG and gelatin. BBG reinforced nanocomposite hydrogels outperformed an empty gelatin hydrogel in terms of mechanical properties. Adding BBG to nanocomposite hydrogels significantly enhanced the bioactivity and durability of composite hydrogels due to strong interactions between BBG and the gelatin network. In vitro findings also revealed that G/BBG nanocomposite hydrogels aided in the formation of apatite. The findings of this study suggest that nanocomposite hydrogels may be valuable biomaterials for bone regeneration.

## Figures and Tables

**Figure 1 polymers-13-02014-f001:**
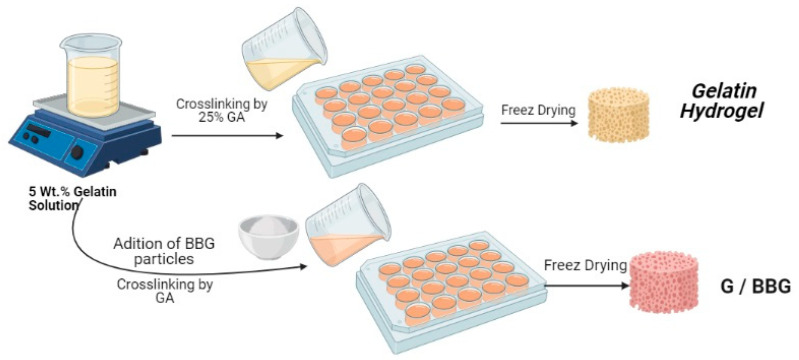
Detailed preparation method of gelatin/borosilicate bioactive glass nanocomposites.

**Figure 2 polymers-13-02014-f002:**
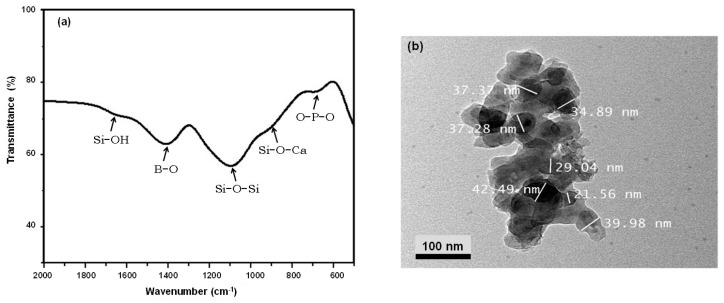
Characterization of BBG nanoparticles: (**a**) FT-IR spectrum and (**b**) TEM micrograph.

**Figure 3 polymers-13-02014-f003:**
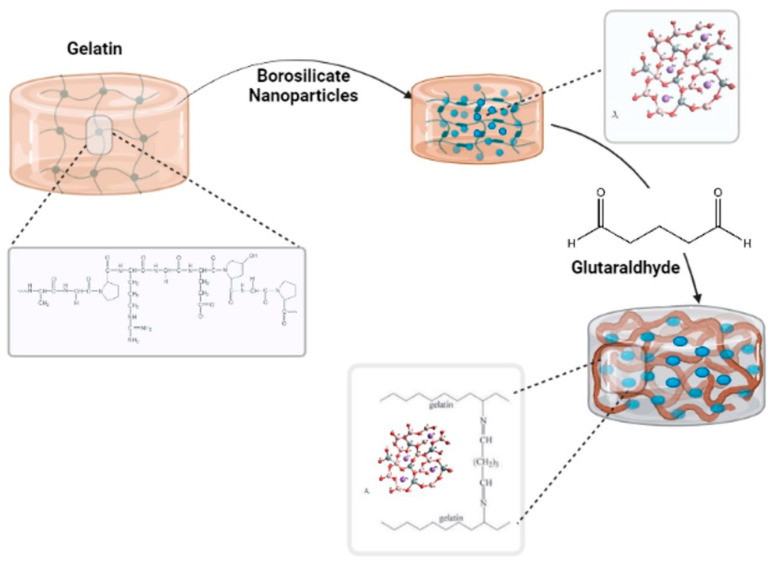
Schematic study of the cross-linking mechanism of gelatin/borosilicate/glutaraldehyde nanocomposite hydrogels.

**Figure 4 polymers-13-02014-f004:**
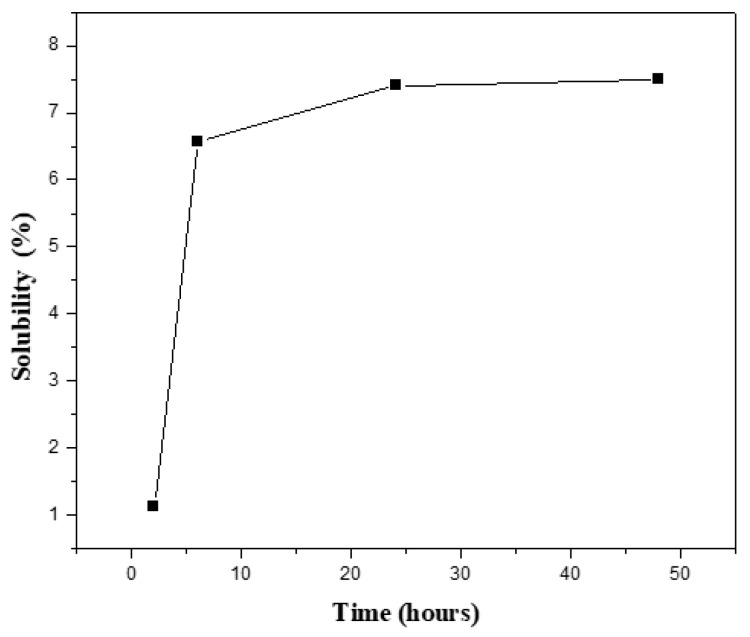
Solubility percentage of the prepared G/5BBG hydrogel nanocomposite with different time intervals.

**Figure 5 polymers-13-02014-f005:**
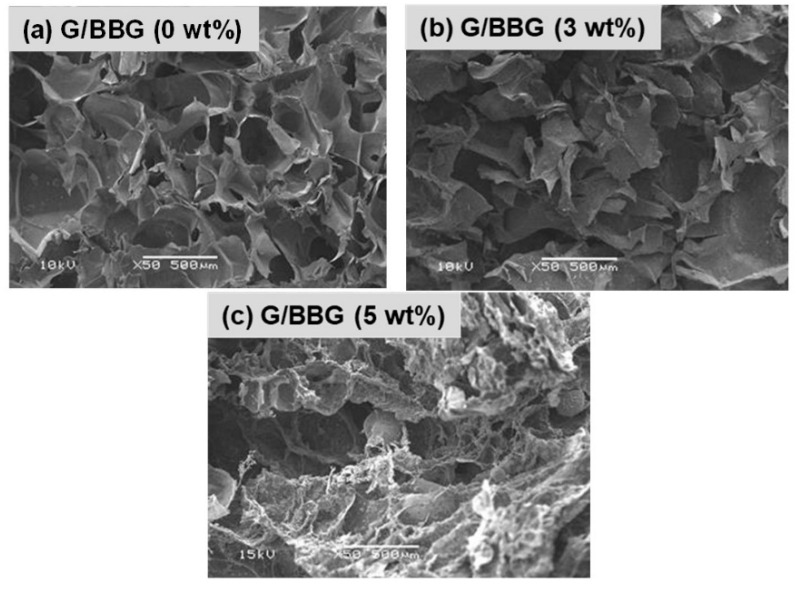
SEM micrographs of the (**a**) G/0 BBG, (**b**) G/3 BBG, (**c**) G/5 BBG nanocomposite hydrogels loaded with different contents of BBG nanoparticles.

**Figure 6 polymers-13-02014-f006:**
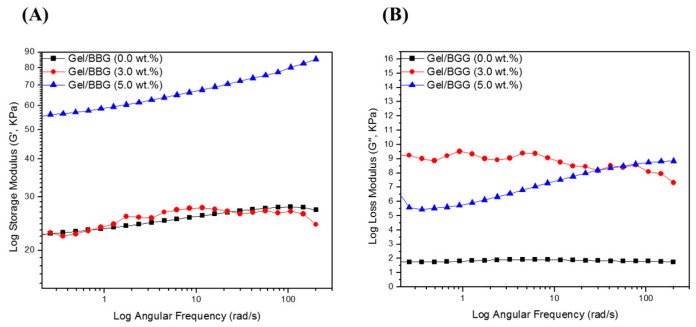
Log scale of (**A**) Storage modulus (Gʹ) and (**B**) loss modulus (Gʺ) of the B/BBG nanocomposite hydrogels as a function of the angular.

**Figure 7 polymers-13-02014-f007:**
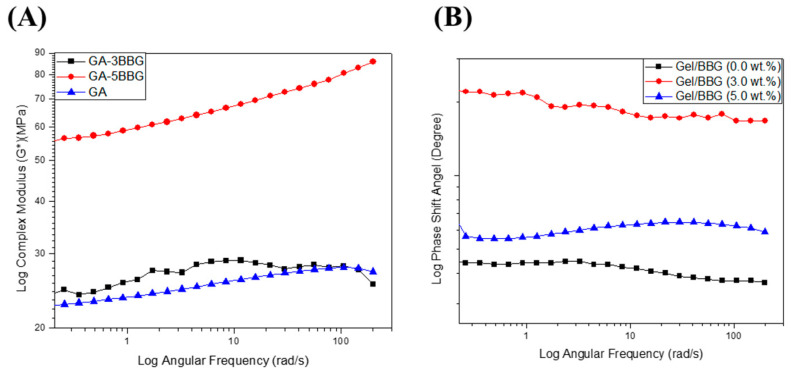
Log scale of (**A**) complex shear modulus (G*) and (**B**) phase shift angle of the B/BBG nanocomposite hydrogels as a function of the angular frequency.

**Figure 8 polymers-13-02014-f008:**
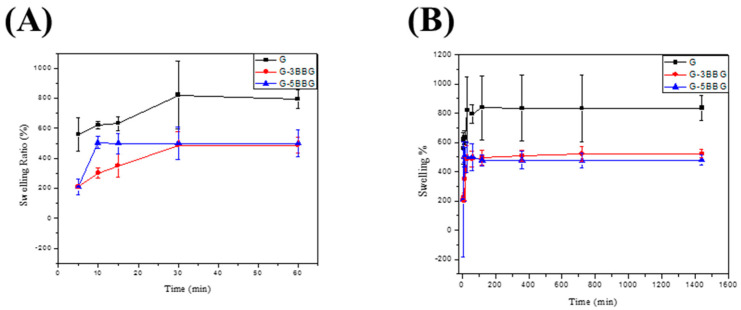
The profile of swelling capacity versus time of the hydrogel composites in deionized water. In each case, the swelling capacity is measured after (**A**) 5, 10, 15, 30, 60 min, (**B**) 5 min, 10 min, 15 min, 30 min, 60 min, 2 h, 6 h, 12 h, and 24 h. The error bar represents the standard deviation based on at least three different measurements performed by three different experimenters.

**Figure 9 polymers-13-02014-f009:**
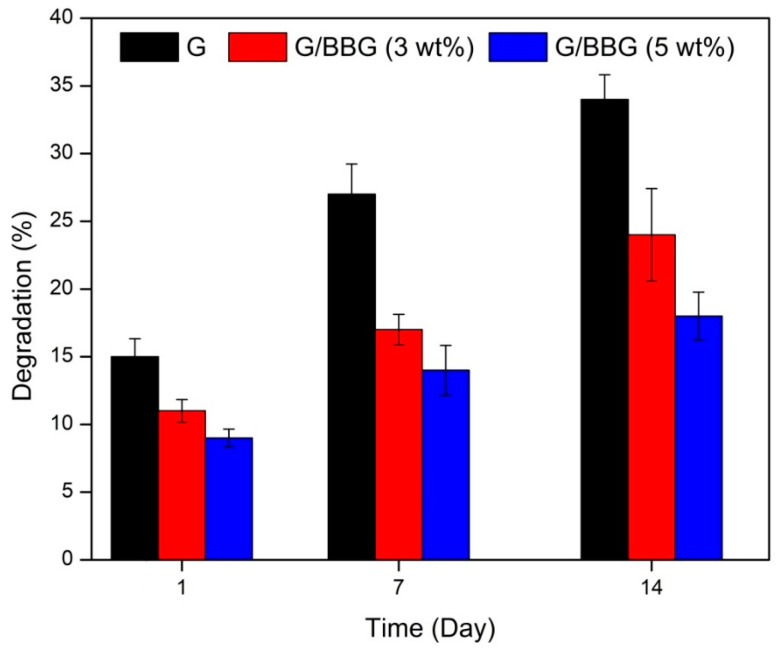
In vitro degradation of the G/BBG nanocomposite hydrogels in PBS at 37 °C.

**Figure 10 polymers-13-02014-f010:**
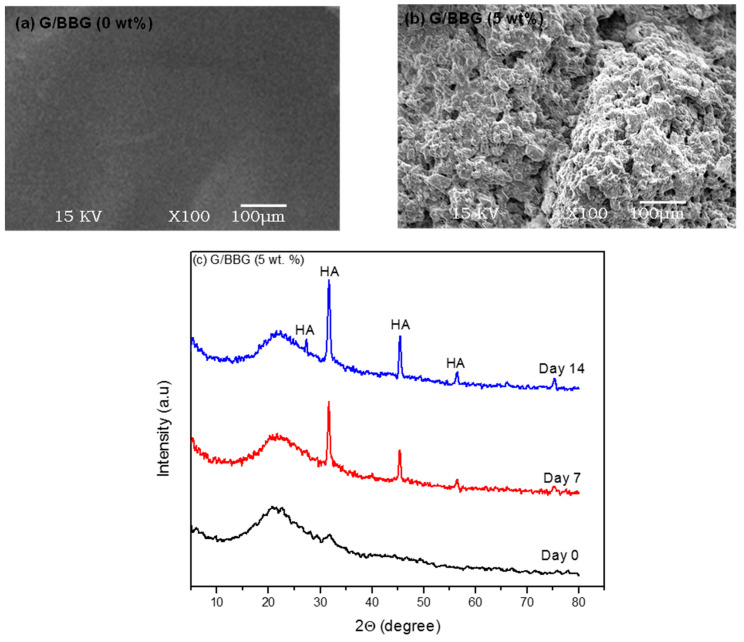
The effect of BBG on the apatite-forming ability of G/BBG nanocomposite hydrogels: (**a**) and (**b**) SEM images showed the better apatite-forming ability after 14 days with the addition of 5 wt.% BBG and (**c**) XRD patterns indicated the HA formation.

**Table 1 polymers-13-02014-t001:** The order of chemicals for SBF preparation.

Order	Reagent	Amount (g)
1	NaCl	12.052
2	NaHCO_3_	0.535
3	KCl	0.337
4	K_2_HPO_4_.3H_2_O	0.346
5	MgCl_2_.6H_2_O	0.477
6	1.0 M-HCl	58.5 mL
7	CaCl_2_	0.438
8	Na_2_SO_4_	0.108
9	(Tris (hydroxymethyl) aminomethane)	9.177
10	1.0 M-HCl	(0–7.5) mL

**Table 2 polymers-13-02014-t002:** Mean values and standard deviations for compressive modulus and compressive strength of the gelatin nanocomposites with different concentrations of borosilicate.

Sample	Compressive Modulus	Compressive Strength
**G/BBG (0 wt.%)**	33.84 ± 1.35	12.20 ± 1.52
**G/BBG (3 wt.%)**	88.52 ± 2.05	12.53 ± 2.16
**G/BBG (5 wt.%)**	173.63 ± 2.82	14.75 ± 1.10

## Data Availability

Data is contained within the article. Samples of the compounds are available from the authors.

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
