# Peer review of "Viscoelasticity, Mechanical Properties, and In Vitro Bioactivity of Gelatin/Borosilicate Bioactive Glass Nanocomposite Hydrogels as Potential Scaffolds for Bone Regeneration"

_polymers, 2021, doi:10.3390/polym13122014_

Round 1
Reviewer 1 Report
The paper describes the preparation of gelatin/borosilicate bioactive glass nanocomposite hydrogels and their testing for viscoelasticity and mechanical properties. In vitro studies were also performed to evaluate their potential for the design of scaffolds for bone regeneration.
The results are proper for publishing in this journal, however, there are issues that must be solved before its recommendation for publication, such as:
- line 31-36 Could you provide recent statistics? It would be helpful to state the urgent need.
- 62-65- Add recent studies-literature survey that use BBG for bone regeneration; previous studies of other hydrogel-composite scaffolds with BBG should be described-advantages, disadvantages.
-line 106-0.5 wt percent GA/conclusions-chemical crosslinking-it is not specified at Materials section; I believe that the crosslinker is glutaraldehyde. The mehod for crosslinkimg is not described; how it was applied...did you immersed the scaffold in the GA solution? Or did you added GA solution. How much? Respective to what? These aspects should be specified. A schematic mechanism should be added as you reffer to hydrogen bonds formes between polymeric chains and the inorganic filler.
-Please, provide some details of the obtained samples. I do not see any reaction yield, crsslinking degree; how the inorganic filler affected these parameters? Did you washed the samples after the synthesis? The samples should have been washed after synthesis to remove the uncrosslinked gelatine. The crosslinking density should be provided.
-For swelling studies and degradation studies...as for all the other tests, the washed samples should have been used. The degradation studies may point the unwashed gelatin removal rather than the degradation ot gelatin.
- Samples (n=6) were equilibrated in PBS for 1 h at 37 °C to test the swelling activity of composite hydrogels. The samples weren’t equilibrated at all, maybe kept in PBS for 1 h. These kind of hydrogels reach equilibrium swelling in 1-2 days or more.
- Line-159. Please, specify the recipe for SBF.
-SEM images were perfomed on the fracture of the specimens? These should be provided in order to observe the inner porous structure of the hydrogels.
Therefore I would suggest publication of the paper after the major revisions are taken into consideration.
With respect,
Author Response
Comments and Suggestions for Authors
We would like to thank you for your comments. We have addressed the comments in the attached word file point by point as well as highlighted in the text itself.
- line 31-36 Could you provide recent statistics? It would be helpful to state the urgent need.
Reply:
This year, 2021, the Transplantation Division (DoT) of the Health Resources and Services Administration (HRSA) is monitoring the effect of the coronavirus (COVID-19), public health emergency on organ donations, and transplantation. On behalf of the Board of Directors, the USA Executive Committee has taken a range of steps to help members document COVID-19 problems surrounding organ donation and transplantation and to help members concentrate the necessary resources on critical clinical services (https://optn.transplant.hrsa.gov) in the short term. Under these conditions, researchers are keen to find new ways to compensate for this organ shortage [1]. - 62-65- Add recent studies-literature survey that uses BBG for bone regeneration; previous studies of other hydrogel-composite scaffolds with BBG should be described-advantages, disadvantages.
Reply:
Bioactive glasses most commonly used in bone tissue engineering (BTE) consist of a silicate network incorporating sodium, calcium, and phosphorous, but modifications with additional elements such as fluorine, magnesium, strontium, iron, silver, boron, potassium, or zinc. In particular, mesoporous silica nanoparticles are widely used as a delivery reagent because silica possesses favorable chemical properties, thermal stability, and biocompatibility. Furthermore, it has been shown that silica-free borate glasses (Borate BG) have low chemical durability and quickly convert to calcium phosphate (or hydroxyapatite) in physiological media and bond directly to bone in a way comparable to silicate-based 45S5 bioglass. Considering the reported positive effect of the B ion, but also on the basis that these glasses can be heat treated (sintered) at lower temperatures than 45S5 bioglass and do not crystallize, boron-containing silicate glasses are currently being investigated. Borate bioactive glasses were obtained from silicate by substituting B2O3 for the molar concentration of SiO2. Both the Melt Quenching and Sol-gel techniques can also produce inorganic glasses [2], [3].
-line 106-0.5 wt percent GA/conclusions-chemical crosslinking-it is not specified in the Materials section; I believe that the crosslinker is glutaraldehyde. The method for crosslinking is not described; how it was applied...did you immersed the scaffold in the GA solution? Or did you added a GA solution? How much? Respective to what? These aspects should be specified. A schematic mechanism should be added as you refer to hydrogen bonds formes between polymeric chains and the inorganic filler. Did you wash the samples after the synthesis? The samples should have been washed after synthesis to remove the un-crosslinked gelatine. The crosslinking density should be provided. Reply: Glutaraldehyde (GA) has been added to the gelatin solution as shown in the following Figure: After stirring for 2 h, the bioglasses containing hydrogels were cross-linked with 5 μl.ml−1 GA and stirred for 1 h at room temperature. the cross-linked samples were repeatedly washed with DI water to remove the remnant GA, Prior to characterization, the obtained nanocomposite hydrogels were cooled to the temperature of - 25 oC overnight and lyophilized under liquid nitrogen in a freeze drier for 48 h to obtain interconnected porous hydrogels.
In the crosslinking reaction, the aldehyde (-CHO) of GA is able to interact chemically with amino groups (-NH2) on gelatin to form chemical bonds, as depicted on the following Figure. Since it is bi-functional, it links two different gelatin molecules, hence its mode of crosslinking [4]. For both crosslinking mechanism, the particles of borate
bioactive glass (BBG) will be incorporated in the crosslinked hydrogel may be physical or chemical interaction by non-bonding oxygen (NBO) atoms.
Please, provide some details of the obtained samples. I do not see any reaction yield, crosslinking degree; how the inorganic filler affected these parameters?
Reply:
The crosslinking density of hydrogels was related to the solubility ratio of hydrogels
determined by immersing in distilled water at room temperature. After different time
intervals, the soluble or unreacted part of samples was extracted by water, samples
removed and weighed after removing excess water from the surfaces using blotting
paper. The solubility ratio was calculated by using the weight of hydrogel after extraction
(G) and the initial weight of hydrogel (Gi) and expressed in percentage [5]
???????????? % = ((
?
??
) ∗ 100)
After 48 h, the solubility ratio equilibrated at 7.5 % so the crosslinking % is high about 93 %.
The modified samples with borosilicate nanoparticles have been crosslinked than unmodified
samples, so both swelling and solubility ratios have been affected.
0 10 20 30 40 50
1
2
3
4
5
6
7
8
Solubility (%)
Time (hours)
-For swelling studies and degradation studies...as for all the other tests, the washed samples
should have been used. The degradation studies may point the unwashed gelatin removal
rather than the degradation of gelatin.
- Samples (n=6) were equilibrated in PBS for 1 h at 37 °C to test the swelling activity of
composite hydrogels. The samples weren’t equilibrated at all, maybe kept in PBS for 1 h.
This kind of hydrogels reaches equilibrium swelling in 1-2 days or more.
Reply:
We completed the swelling ratios to reach the equilibrium. To evaluate the sensitivity of the
hydrogel by surrounding factors, the swelling ratio was measured in a timely manner. The
swelling behavior of hydrogels is affected by the hydrophilic group present in the polymer chains
[6]. The swelling ratios have been equilibrated after 30 minutes as shown in the following Figure.
0 10 20 30 40 50 60
-200
0
200
400
600
800
1000
Swelling Ratio (%)
Time (min)
G
G-3BBG
G-5BBG
0 200 400 600 800 1000 1200 1400 1600
-200
0
200
400
600
800
1000
1200
Swelling %
Time (min)
G
G-3BBG
G-5BBG
The profile of swelling capacity versus time of the hydrogel powders in deionized water. In each
case, the swelling capacity is measured after 5 min, 10 min, 15 min, 30 min, 1 h, 2 h, and 6 h,
12h, and 24 h. The error bar represents the standard deviation based on at least three different
measurements performed by three different experimenters.
Line-159. Please, specify the recipe for SBF.
Reply:
The preparation of 1 liter of SBF, was as shown in (Table) by order: If necessary, the pH was
adjusted using hydrochloric acid or sodium hydroxide [7]–[9].
Table 2: The order of chemicals for SBF preparation.
Order Reagent Amount (g)
1 NaCl 12.052
2 NaHCO3 0.535
3 KCl 0.337
4 K2HPO4.3H2O 0.346
5 MgCl2.6H2O 0.477
6 1.0 M- HCl 58.5 ml
7 CaCl2 0.438
8 Na2SO4 0.108
(A) (B)
9
(Tris (hydroxymethyl) aminomethane)
9.177
10
1.0M-HCl
(0-7.5) ml -SEM images were performed on the fracture of the specimens? These should be provided in order to observe the inner porous structure of the hydrogels. Reply: Samples were freeze-dried and sputter-coated with gold to a thickness of 400 Å in a sputter-coating unit (JFC 1100 E) prior to imaging by SEM.

Reviewer 2 Report
I have reviewed a manuscript entitled “Viscoelasticity, mechanical properties, and in-vitro bioactivity of gelatin/borosilicate bioactive glass nanocomposite hydrogels as potential scaffolds for bone regeneration”. This work aimed to investigate the hydrogel composed of gelatin and borosilicate bioactive glass. They monitored the changes in the mechanical and morphological behavior of the gelatin hydrogel upon the addition of filler. I think this work can be considered for publication after addressing the following comments:
Comment 1: captions of figures 3 and 4 are mixed!
Comment 2: please redraw figures 3 and 4 on a log-log scale.
Comment 3: please extend the swelling ratio experiment to ensure the samples reached equilibrium.
Author Response
Reviewer 2
Comments and Suggestions for Authors
We would like to thank you for your comments. We have addressed the comments in the
attached word file point by point as well as highlighted in the text itself.
Comment 1: captions of figures 3 and 4 are mixed!
Reply:
The Figures have been separated in the main text.
Comment 2: please redraw figures 3 and 4 on a log-log scale.
Reply:
1 10 100
20
30
40
50
60
70
80
90
Gel/BBG (0.0 wt.%)
Gel/BBG (3.0 wt.%)
Gel/BBG (5.0 wt.%)
Log Storage Modulus (G', KPa)
Log Angular Frequency (rad/s)
1 10 100
0
1
2
3
4
5
6
7
8
9
10
11
12
13
14
15
16 Gel/BGG (0.0 wt.%)
Gel/BGG (3.0 wt.%)
Gel/BGG (5.0 wt.%)
Log Loss Modulus (G'', KPa)
Log Angular Frequency (rad/s)
Figure 3. Log scale of (a) Storage modulus (Gʹ) and (b) loss modulus (Gʺ) of the B/BBG nanocomposite hydrogels as a function
of the angular
(A) (B)
1 10 100
20
30
40
50
60
70
80
90
Log Complex Modulus (G*)(MPa)
Log Angular Frequency (rad/s)
GA-3BBG
GA-5BBG
GA
1 10 100
Log Phase Shift Angel (Degree)
Log Angular Frequency (rad/s)
Gel/BBG (0.0 wt.%)
Gel/BBG (3.0 wt.%)
Gel/BBG (5.0 wt.%)
Figure 4. Log scale of (a) Complex shear modulus (G*) and (b) phase shift angle of the B/BBG nanocomposite hydrogels as a
function of the angular frequency
Comment 3: please extend the swelling ratio experiment to ensure the samples reached
equilibrium.
Reply:
We completed the swelling ratios to reach the equilibrium. To evaluate the sensitivity of the
hydrogel by surrounding factors, the swelling ratio was measured in a timely manner. The
swelling behavior of hydrogels is affected by the hydrophilic group present in the polymer chains
[4]. The swelling ratios has been equilibrated after 30 minutes as shown in the following Figure.
0 10 20 30 40 50 60
-200
0
200
400
600
800
1000
Swelling Ratio (%)
Time (min)
G
G-3BBG
G-5BBG
0 200 400 600 800 1000 1200 1400 1600
-200
0
200
400
600
800
1000
1200
Swelling %
Time (min)
G
G-3BBG
G-5BBG
The profile of swelling capacity versus time of the hydrogel powders in deionized water. In each
case, the swelling capacity is measured after 5 min, 10 min, 15 min, 30 min, 1 h, 2 h, and 6 h,
12h, and 24 h. The error bar represents the standard deviation based on at least three different
measurements performed by three different experimenters
(A) (B)
(A) (B)

Round 2
Reviewer 1 Report
Dear Authors,
The manuscript has been improved and can be published.
With respect,
Author Response
Thank you for your kind revision of our manuscript entitled “Viscoelasticity, mechanical properties, and in-vitro bioactivity of gelatin/borosilicate bioactive glass nanocomposite hydrogels as potential scaffolds for bone regeneration”, Manuscript ID: polymers-1224832. We revised the grammar of the article as a whole.